# Enhancing Image Quality via Robust Noise Filtering Using Redescending M-Estimators

**DOI:** 10.3390/e25081176

**Published:** 2023-08-07

**Authors:** Ángel Arturo Rendón-Castro, Dante Mújica-Vargas, Antonio Luna-Álvarez, Jean Marie Vianney Kinani

**Affiliations:** 1Department of Computer Science, Tecnológico Nacional de México/CENIDET, Interior Internado Palmira S/N, Palmira, Cuernavaca 62490, Mexico; angel.rendon18ce@cenidet.edu.mx (Á.A.R.-C.);; 2Unidad Profesional Interdiciplinaria de Ingeniería Campus Hidalgo, Instituto Politécnico Nacional, Pachuca 07738, Mexico

**Keywords:** noise filtering, redescending M-estimator, image processing, multiplicative noise, additive noise, impulsive noise

## Abstract

In the field of image processing, noise represents an unwanted component that can occur during signal acquisition, transmission, and storage. In this paper, we introduce an efficient method that incorporates redescending M-estimators within the framework of Wiener estimation. The proposed approach effectively suppresses impulsive, additive, and multiplicative noise across varied densities. Our proposed filter operates on both grayscale and color images; it uses local information obtained from the Wiener filter and robust outlier rejection based on Insha and Hampel’s tripartite redescending influence functions. The effectiveness of the proposed method is verified through qualitative and quantitative results, using metrics such as PSNR, MAE, and SSIM.

## 1. Introduction

Crucial tasks, such as image processing and medical diagnostics through imaging, require the absence of noise and acceptable quality. This necessitates the suppression of noise without deteriorating fine details or suppressing essential data. Image acquisition, storage, and transmission often introduce various types of noise. For most images, the noise mainly comes from additive, multiplicative, and impulsive noise [1]. Additive noise degrades the image due to pixel intensity fluctuations in the 2D space, with each pixel’s degradation characterized by a Gaussian-like distribution (Gaussian noise). Multiplicative noise, also known as speckle noise, is usually generated when images of complex objects are acquired using highly complex waves; it depends on the signal that produces it and it is difficult to eliminate with traditional noise models. Impulsive noise could be modeled by employing fixed intensity pixel values (salt and pepper noise) where some image pixels are altered to 0 or 255 [2,3,4]. This paper proposes a filter designed to process additive, multiplicative, and impulsive noise using redescending M-estimators. The robust functions implemented are Insha and Hampel’s three-part redescending functions, implemented in the context of a Wiener filter due to their advantages in local measures. The proposed work is assessed using PSNR, SSIM, and MAE metrics and compared with four prevalent methods to process the three types of noise.

In this research, our primary motivation for developing this filter using redescending M-estimators stems from the understanding that additive, impulsive, and multiplicative noise are dominant sources of image corruption. Such noise can emerge during image capture and transmission because of uncontrollable environmental factors or sensor-related characteristics. Furthermore, impulsive noise often manifests during image storage and manipulation. Considering these factors collectively, we believe that our proposed application has broad applicability across various fields, with particular significance in the realm of medical imaging. By mitigating the impact of noise, our solution could assist experts in accurately diagnosing potential pathologies in corrupted images influenced by the aforementioned noise types.

The contribution of this research consists of introducing a noise suppression filter based on redescending M-estimators. The filter is supported by robust processing through the theory of redescending M-estimators, which makes it more tolerant to high noise densities. We compare the proposed filter with four popular filters currently used for general noise suppression. The performance of the filter is evaluated through different experiments, using metrics such as PSNR, SSIM, and MAE.

This paper is organized as follows: In Section 2, we introduce the basic concepts required to understand the work. Section 3 presents the proposed denoising filter and its mathematical background. In Section 4, we present the experimental results and a comparative analysis of the image restoration quality using metrics in comparison to other current methods in the literature. Finally, in Section 5, we provide a synopsis of the main results and recommendations for future work.

## 2. Related Work

Several methods have been developed to address the removal of one or a combination of the three types of noise. For the purposes of this research, these methods can be classified into five major classes [5,6,7]:

1. Additive and multiplicative filtering: This class includes widely used non-local means (NLM)-based filters [8], which estimate the value of a particular pixel by considering information from the entire image, preserving relevant details. It also includes anisotropic diffusion (AD)-based filters [9,10], which adaptively apply a diffusion process based on the local structure of the image. In [9], an anisotropic diffusion coefficient with an image-dependent threshold parameter was proposed for low densities of Gaussian and speckle noise. While in [10], a method for only suppressing speckle noise was proposed, which combines the use of a Canny operator to enhance an anisotropic diffusion equation. Although these filters are good at preserving details, they fail to handle high noise densities and are not robust against impulse noise. Additionally, bilateral-based filters [11,12,13,14] consider both spatial proximity and intensity similarity between pixels, ensuring that nearby pixels have a significant influence on the filtering process. In [11,12], they focus on universal noise suppressors, but they only process Gaussian and impulse noise. Meanwhile, in [13,14], the authors address the optimization and adaptation of the bilateral algorithm to work with color images. This approach, while effective in terms of parameter flexibility and preserving certain edges in images, lacks handling high noise densities, processing multiplicative noise, and certain blurring of details. Another approach involves estimating the noise level by transforming an image into other domains. For example, in [15], a local complexity estimation in the wavelet domain was proposed for MRI denoising. In [16], a collaborative 3D domain transformation was introduced for noise removal. Additionally, in [17], a hidden Markov model was proposed for transforming an image into other spaces and suppressing Gaussian noise. This type of work often leverages the increase/change of information through domain transformation, but image details and noise cannot be completely separated, and certain image characteristics can be lost. One general drawback of the mentioned filters is that they can introduce a certain degree of blurriness in the image. This is due to considering information from the entire image or applying adaptive diffusion processes, which may result in the loss of fine details or the blurring of important edges. This can impact the visual quality and accuracy of the filtered image, especially in areas with sharp intensity or texture changes. Additionally, some of these filters cannot handle high noise densities.

2. Impulsive filtering: This class primarily focuses on nonlinear filters and is commonly addressed through median-based filters, as in [18], where a fuzzy paradigm was also applied. Similarly, iterative mean/median filters were used in the works of Chen [19,20], and robust statistical methods that achieve remarkable results at rejecting atypical data [21], which is a distinctive characteristic of impulsive noise. These filters typically perform well with high noise densities but are limited to only one type of noise, and their computational costs can be exhaustive due to their iterative processing.

3. Neural networks for noise removal: This category encompasses various approaches. For instance, in [22], a model based on deep convolutional neural networks (CNNs) was proposed for removing multiplicative noise. In [23], a complex-valued CNN was applied for Gaussian denoising. In [24], two deep CNNs were combined to leverage more features for image denoising, and batch renormalization was used to address the small mini-batch problem. Furthermore, in [25], Tian et al. introduced a deep CNN with batch renormalization. Additionally, in [26], the same authors worked with a three-stage CNN by incorporating the wavelet transform, which is highly effective for denoising noisy images corrupted by unknown noise. While these methods can achieve excellent results, they often require substantial resources for designing and training, high computational power, overfitting risks, difficulty in internal interpretation, and sensitivity to the quality and quantity of training data. It is worth noting that these approaches typically rely on extensive training databases, although some methods, like Self2self [27], only use a single image for training. However, their training processes can be very slow.

4. Sparse and low-rank models: These models focus on decomposing a matrix into two components, i.e., a low-rank component and a sparse component. The low-rank component can be approximated as the sum of a relatively small number of column vectors, while the “sparse” component refers to the property of having many zero or near-zero elements. These models are achieved through the minimization of an objective function that penalizes the difference between the original matrix and the sum of the low-rank and sparse matrices. In [28,29,30,31], they were applied to hyperspectral images with multiple channels. While these models can work for various types of noise and handle large volumes of data, they also have some limitations. They can be computationally intensive due to the decomposition and reconstruction of the matrices. Additionally, they are sensitive to parameter selection, require appropriate training data, and have limitations in their ability to handle certain types of noise.

5. Robust estimation for filtering: This category involves the application of robust estimation techniques to eliminate impulsive noise. Previous studies have utilized M-estimators. In [32], a median redescending estimator was employed for impulsive denoising in grayscale images. Furthermore, in [33], we extended the median redescending M-estimator for color images, multi-core processing, and random-value impulsive noise. These robust estimations utilize redescending influence functions (ψ) for effective impulsive noise elimination. Another approach involves an NLM (non-local means) method for suppressing Gaussian noise using Tukey’s biweight estimation [34].

Therefore, noise suppression represents a challenging task that can be approached from different perspectives. However, the approaches discussed so far often focus on low noise densities and lack the ability to handle all three types of noise. For this reason, in this research, we aim to develop a filter that does not require training and can effectively handle different types of noise, even at high densities. While the primary focus of this research is to contribute to the diversity of approaches in image noise removal methods, we strongly believe in addressing current and future challenges and exploring efficient and adaptable solutions across diverse contexts and applications. Algorithms that go beyond the classes presented in this section offer unique advantages and can serve as valuable complements to enrich the overall research field.

## 3. Principles

### 3.1. Noise Models

Noise is considered an unwanted component in signal processing, and it can occur during the capturing, processing, and storage of information [35]. If *f* is considered a signal, it can be decomposed into a desired component *g* and an undesired component representing noise *q*. The most common types of noise are additive noise (f=g+q), multiplicative noise (f=gq), and impulsive noise (f=g(q)). These are the types of noise addressed in this work, and their characteristics are summarized in Table 1.

### 3.2. M-Estimators

The maximum likelihood estimator proposed by Huber in 1964 has been generalized to M estimators, offering reasonable bias and flexibility by treating the mean and median as special cases [37]. M estimators, denoted as $T_{n}=T_{n}\left(x_{1},\ldots ,x_{n}\right)$, minimize the objective function [38,39,40]: (1)$$\sum_{i=1}^{n}\rho \left(x_{i},T_{n}\right)\to \min$$

Here, $T_{n}$ represents the desired estimate, and ρ is the loss function. The minimization problem can also be expressed as follows: (2)$$\sum_{i=1}^{n}\psi \left(x_{i},T_{n}\right)$$
where $\psi (x,T_{n})$ is the influence function, defined as $\psi \left(x,T_{n}\right)=\frac{\partial }{\partial T_{n}}\rho \left(x,T_{n}\right)$. The loss function ρ is assumed to be symmetric and positive definite; it has a unique minimum at zero, and possesses a partial derivative.

### 3.3. Redescending M-Estimators

A particular type of M-estimator can completely reject extreme values, which implies that their ψ functions decay away from the central region and do not decay near the origin. These are called redescending M-estimators and are defined as [41]: (3)$$\psi_{r}\left(x\right)=\left\{\psi \left(x\right)=0\;\forall \;|x|\geq r\right\}$$
where 0<r<∞ is the threshold that allows the limits of the influence function to be set. Redescending estimators are designed to vanish outside the central region and settle to zero if the threshold *r* is reached. Their efficiency is due to function psi having a high break point and does not entirely ignore moderately large outliers [42]. The estimators used are Hampel’s three-part redescending M-estimator and Insha’s redescending M-estimator, which are discussed below.

#### 3.3.1. Hampel’s Three-Part Redescending

The Hampel M-estimators maintain a strategic approach to address outliers by delineating regions that mirror the effects of outliers to varying extents. The Hampel three-part redescending estimator is the most suitable for mitigating the influence of outliers. In contrast, other M-estimators, such as Huber’s and Tukey’s, do not provide the requisite accuracy to ensure an adequate level of precision when confronting specific outlier types. The function of Hampel’s three-part redescending estimator, seen as a robust measure, can be interpreted as a combination of the $l_{2}$ norm and the $l_{1}$ norm, excluding outliers; its function ψ is given by:    
(4)$$\psi_{HAM_{\left(a,b,r\right)}}\left(x_{i,j}\right)=\left\{\begin{array}{cc} x & 0\leq |x|\leq a\\ a\cdot sgn(x) & a\leq |x|\leq r\\ a\cdot \frac{r-|x|}{r-b} & b\leq |x|\leq r\\ 0 & 0\leq |x| \end{array}\right.$$

The ψ representation of Hampel’s three-part redescending function is shown in Figure 1.

#### 3.3.2. Insha

This estimator covers some drawbacks of the redescending M-estimators and is considered a tool for outlier detection and robust regression [43]. The form and properties of its corresponding ψ function are discussed below: (5)$$\rho \left(x\right)=\frac{c^{2}}{4}\left[arctan{\left(\frac{x}{c}\right)}^{2}+\frac{c^{2}+x^{2}}{c^{4}+x^{4}}\right]\;for\;\left|x\right|\geq 0$$
where *c* is the fit constant; for the *i*-th observation, variables $x_{i}$ are the residuals scaled over MAD. With respect to *x*, the following ψ function is obtained: (6)$$\psi_{I_{\left(r\right)}}\left(x\right)=x\cdot {\left[1+{\left(\frac{x}{c}\right)}^{4}\right]}^{2}\;for\;\left|x\right|\geq 0.$$

The graph of the ψ Insha function is shown in Figure 2.

## 4. Proposed Noise Suppression Filter

Our proposal focuses on eliminating additive and multiplicative noise with the Insha estimator because the normal distribution of these noises can be optimally modeled through this approach. For impulsive noise, Hampel’s three-part estimator is used, which has an adaptive behavior by the MAD component that allows, through its thresholds, the correct suppression of impulsive noise. To make the process more robust, we follow the structure of the Wiener filter. Wiener filtering minimizes the overall mean square error in the inverse filtering and noise smoothing process. Wiener estimates the local mean and variance around each pixel [44].
(7)$$\mu =\frac{1}{NM}\sum_{n_{1},n_{2}\in \eta }a\left(n_{1},n_{2}\right),$$
and
(8)$$\sigma^{2}=\frac{1}{NM}\sum_{n_{1},n_{2}\in \eta }a^{2}\left(n_{1},n_{2}\right)-\mu^{2},$$
where η is the local N×M neighborhood of each pixel in image *X*. Wiener creates a per-pixel Wiener filter using these estimates. Using the above measurements, we have the following equation:(9)$$b\left(n_{1},n_{2}\right)=\mu +\frac{\sigma^{2}}{v^{2}}\left(a\left(n_{1},n_{2}\right)-\mu \right).$$
where $v^{2}$ is the variance of the noise. If the noise variance is not given, Wiener uses the average of all estimated local variances. Then, considering the above, we rewrite the local means in terms of a redescending estimator:(10)$$\mu \left(\hat{x}\right)=\frac{1}{nm}\sum_{i,j}\psi \left(x\right)$$
where $\tilde{x}$ is the observation of the noisy image and $\hat{x}$ is the result of the component of the Wiener filter. The local variance is as follows:(11)$$\varsigma^{2}\left(\hat{x}\right)=\frac{1}{nm}\sum_{i,j\in \eta }\psi^{2}\left(x_{ij}\right)-\mu^{2}\left(x\right)$$ The noise variance is as follows: (12)$$v^{2}\left(\hat{x}\right)=k\cdot med\{|\psi x_{ij}-med\left(\psi \left(x_{\eta }\right)\right)\left|\right\}$$
where *k* is a scaling factor for normally distributed data; it uses the reciprocal of the quantile function $\Phi^{-1}$, while 3/4 represents the portion that covers 50% (between 1/4 and 3/4) of the standard normal cumulative distribution function:(13)$$k=\frac{1}{\Phi^{-1}\frac{3}{4}}=1.4826$$

As a result of the above equations, using a filter with the Wiener structure but operating with the properties of the redescending M-estimator, we have the following: (14)$$\tilde{x}=\psi \left(x\right)+\frac{\varsigma^{2}\left(\hat{x}\right)-v^{2}\left(\hat{x}\right)}{\varsigma^{2}\left(\hat{x}\right)}\cdot \left(x-\psi \left(x\right)\right)$$ The influence function ψ can take the value of Insha = $\psi_{I_{\left(r\right)}}$ or Hampel = $\psi_{HAM_{\left(a,b,r\right)}}$, depending on the noise to be processed. Thus, we combine the two non-parametric methods that allow the elimination of impulsive and multiplicative noise by utilizing a Wiener smoothing procedure and the robustness provided by a redescending M-estimator. The influence functions used are shown in Table 2.

### Proposed Algorithm

The basic structure of the filtering process can be seen in the pseudocode Algorithm 1.
**Algorithm 1** Robust redescending M-estimator filter.**Require:** Noisy image X in grayscale with size N×M.**Ensure:** Select an influence function of Table 2, depending on the type of noise.  1:**for**i=1 to *M* **do**    2:    **for** j=1 to *N* **do**    3:        $\mu \left(\hat{x}\right)\leftarrow \frac{1}{nm}\sum_{i,j}\psi \left(x\right)$                ▹ Compute local mean.    4:        $\varsigma^{2}\left(\hat{x}\right)\leftarrow \frac{1}{nm}\sum_{i,j\in \eta }\psi \left(x^{2}-\mu^{2}\left(x\right)\right)$        ▹ Compute local variance.    5:        $v^{2}\left(\hat{x}\right)=1.4823\cdot med\{|\psi x-med\left(\psi \left(x\right)\right)\left|\right\}$   ▹ Compute noise variance.    6:        $\tilde{x}=\psi \left(x\right)+\frac{\varsigma^{2}\left(\hat{x}\right)-v^{2}\left(\hat{x}\right)}{\varsigma^{2}\left(\hat{x}\right)}\cdot \left(x-\psi \left(x\right)\right)$     ▹ Redescending M-estimator.    7:        $Y_{i,j}\leftarrow \tilde{x}$       ▹ The calculated value is written in the output image.    8:    **end for**   9:**end for**     **Output**: Filtered image *Y* with size M×N.

## 5. Experimentation and Results

The performance of the proposed filter is validated through three tests: standard and medical grayscale image processing, image size and batch processing, color image and video processing.

### 5.1. Standard and Medical Grayscale Image Processing

In the first instance, this experiment was carried out using quality metrics and test images, such as the standard Lena (512×512 size), lbox_66720-Afbeelding7 (437×520 size and abbreviated as e2), mdb332DNORM (425×390 size and abbreviated as e3), and 00000152_016 (272×530 size and abbreviated as e4) from the datasets [45,46,47]. In the execution, images were corrupted by additive and multiplicative noise from 0.02 to 0.12 in variance, and impulsive from 0.1 to 0.6. The performance of our filter was compared with four state-of-the-art filtering methods that were highly tested to suppress different kinds of noise, such as non-local means (NLM) [48,49], BM3D [16], bilateral filter [14], trilateral filter [11], and estimate parameters for anisotropic diffusion filtering [50,51]. We programmed the proposed filter in MATLAB via an Intel(R) Core(TM) processor i5-8400 CPU @2.8 GHz with six cores and 32 GB of RAM. The comparative methods for ease were also implemented in MATLAB. The metrics used were the following.

The noise suppressing quality was quantified by using the peak signal-to-noise ratio (PSNR) [52] and structural similarity index (SSIM) [53]; the mean absolute error (MAE) was used to quantify the preservation of the fine details of the image [52]. These three metrics can be determined from the following expressions: (15)$$\mathrm{PSNR}=10\cdot log\left[\frac{{\left(max\left(x\left(i,j\right)\right)\right)}^{2}}{MSE}\right],$$
where MSE is the mean squared error and is determined by: (16)$$\mathrm{MSE}=\frac{1}{M\cdot N}\sum_{i=1}^{M}\sum_{j=1}^{N}{\left[x\left(i,j\right)-\hat{e}\left(i,j\right)\right]}^{2},$$
where M·N represents the image sizes that are analyzed, x(i,j) is the original image, and $\hat{e}\left(i,j\right)$ is the improved image. The SSIM metric, in a simplified form, is calculated by the following expression: (17)$$\mathrm{SSIM}(x,y)=\frac{\left(2\mu_{x}\mu_{y}+C_{1}\right)\cdot \left(2\sigma_{xy}+C_{2}\right)}{\left(\mu_{x}^{2}\mu_{y}^{2}+C_{1}\right)\cdot \left(\sigma_{x}^{2}+\sigma_{y}^{2}+C_{2}\right)},$$
where *x* is the original image, *y* is the refined image, $\mu_{x}$ and $\mu_{y}$ are the luminance values, $\sigma_{x}$ and $\sigma_{y}$ are the contrast values, and $C_{1}$ and $C_{2}$ are two constant values. On the other hand, MAE can be computed by: (18)$$\mathrm{MAE}=\frac{1}{M\cdot N}\sum_{i=1}^{M}\sum_{j=1}^{N}\left[x\left(i,j\right)-\hat{e}\left(i,j\right)\right],$$

The results of experimentation with the PSNR, SSIM, and MAE metrics can be observed in Table 3 for the Lena grayscale image. According to the table, we can see that the proposed filter—considering the redescending influence functions at the lowest densities—has a lower-medium performance. However, as the noise density increases, the performance of our proposed filter is more efficient than the comparative filters, above all, for multiplicative noise.

The results can be more easily identified in the graphs of Figure 3. These graphs show the average of the metrics for the four test images using the Insha estimator for additive and multiplicative noise, exhibiting a more tolerant behavior with increasing noise density. On the other hand, for the Hampel estimator, an inverse behavior is observed, as it has a more efficient performance at low densities of impulsive noise. However, as the noise density increases, it becomes more affected.

Figure 4 confirms that the filtered Lena image using redescending functions aligns with the expected metric results. Improved performance is observed at high noise densities for both additive and multiplicative noise with the Insha estimator, while visually, the processed images exhibit better preservation of details. In contrast, the Hampel estimator shows superior results for impulsive noise at low densities.

Figure 5 presents the results for images e2, e3, and e4, demonstrating the effectiveness of the proposed filter in efficiently removing various types of noise from medical images.

### 5.2. Image Size and Batch Processing

To assess the algorithm’s complexity, the execution times of various algorithms were measured using the resized grayscale Lena image. The results of this experiment are shown in Table 4. The results indicate that the proposed approach has a reasonable execution time compared to the comparative methods. However, it is worth noting that utilizing a GPU is recommended to further accelerate the execution time of the proposed method.

In general, it can be said that the proposed algorithm has a complexity of $O(n^{2})$ since it iterates through all the pixels of the image, and for each pixel, a specific calculation is performed depending on the type of noise. However, for the multiplicative and impulsive noise types, better results are usually obtained compared to the compared algorithms.

To validate the effectiveness of our proposed approach, it was implemented on various datasets to assess its performance across different densities of the three types of noise. The datasets used included the mammographic image analysis society digital mammogram database (MIAS) [46], comprising one hundred images, twelve standard grayscale images from BSD68 [54], and fifty images from the dataset of breast ultrasound images (DBUI) [55]. The results obtained for different noise densities demonstrated satisfactory performance across different image types. The results of the highest densities of each noise type are presented in Table 5.

### 5.3. Color Image and Video Processing

The necessary adjustments were made to implement the filtering algorithms for color images, specifically for images in the RGB color space (red, green, and blue). These adjustments focused on separating the images into their three RGB channels, processing each channel separately, and then recombining the images at the end. From each image, three copies were created with impulsive, additive, and multiplicative noise, respectively. The impulsive noise copies were processed using the Hampel estimator, while the images with additive and multiplicative noise were processed using the Insha estimator. In addition to the proposed filtering algorithms, non-local means (NLM), anisotropic diffusion (AD), bilateral, block-matching, and 3D filtering (BM3D) algorithms were also implemented in the images. Thus, 108 resulting images were obtained from each image.

The fact that three channels of information are now being processed has resulted in an increase in information, which in turn has led to improved performance for the proposed work. For high-density noise experimentation, the standard color images, i.e., Lena, Baboon, Goldhill, Boats, Barbara, and Peppers, were used, along with a positron emission tomography (PET) image. Quantitative results for color image filtering can be seen in Figure 6 for the baboon image and in Figure 7 for brain PET. One can observe how the increase in information benefits the proposed filter, especially for impulsive noise.

An experiment was conducted with algorithms using the color image datasets Kodak24 [56] and CBSD68 [54] at the highest levels of density used in this work. The average results of the three types of noise can be observed in the graphs of Figure 8, where the proposed filter performs better in the SSIM and MAE metrics, and second best in the PSNR metric.

Another additional objective involved processing large batches of images, particularly in videos. For this implementation, a corrupted ultrasound video was used. The formulated NLM and Wiener estimators and algorithms were implemented. The video consists of 687 frames, each with dimensions of 540x360 pixels. Some of these frames can be seen in Figure 9.

To evaluate this implementation, a no-reference image quality metric called the naturalness image quality evaluator (NIQE) was used. NIQE operates exclusively by measuring quantifiable deviations from statistical patterns observed in natural images, without relying on any prior knowledge or information.

In Figure 10, the proposed filter achieved a lower value compared to the others in the proposed work. Having a low value means that the image quality is perceived as good. The NIQE metric is used to evaluate the visual quality of images and provides a numerical score, indicating how close an image is to being perceived as natural by humans. Therefore, a low value in the NIQE metric indicates that the image has good visual quality.

## 6. Conclusions

This work proposes a robust filter for noise suppression based on redescending M-estimators. The aim is to provide a filter that could restore images from additive, multiplicative and impulsive noise. An application in the medical image field could help experts diagnose possible pathologies better. We propose using two influence functions. The first is the Insha; due to its approximation to a normal distribution, it allows the elimination of additive and multiplicative noise. The second, Hampel’s three-part redescending, was used to suppress impulsive noise. The proposed filter shows better performances compared to MLN and AD filters in terms of PSNR, SSIM, and MAE for high additive and multiplicative noise densities, while for the impulsive noise, the best results were at low densities. Considering the processing of RGB images, positive results were obtained according to the metrics, partly due to the information gain resulting from the three channels. Additionally, in the video processing experiment, a good result was indicated through the NIQE metric. In future work, we will consider that the functions can be improved through iterative processes, implementing a noise detector, and incorporating additional or different robust functions. However, it is important to note that this may come with high computational costs, so we could also focus on exploring parallel processing.

## Figures and Tables

**Figure 1 entropy-25-01176-f001:**
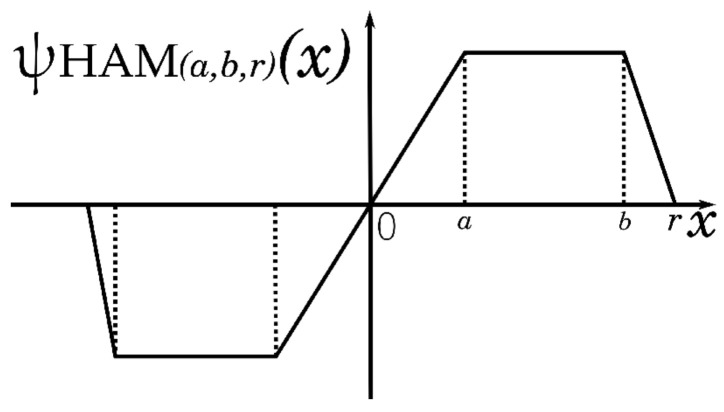
Graph of Hampel’s three-part redescending function.

**Figure 2 entropy-25-01176-f002:**
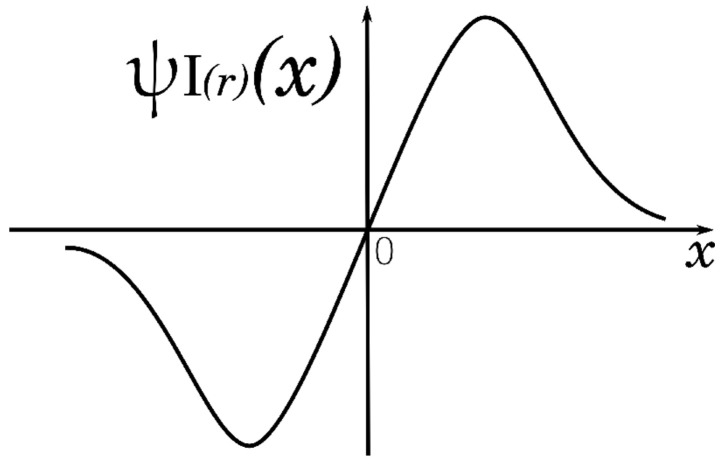
Graph of the Insha function.

**Figure 3 entropy-25-01176-f003:**
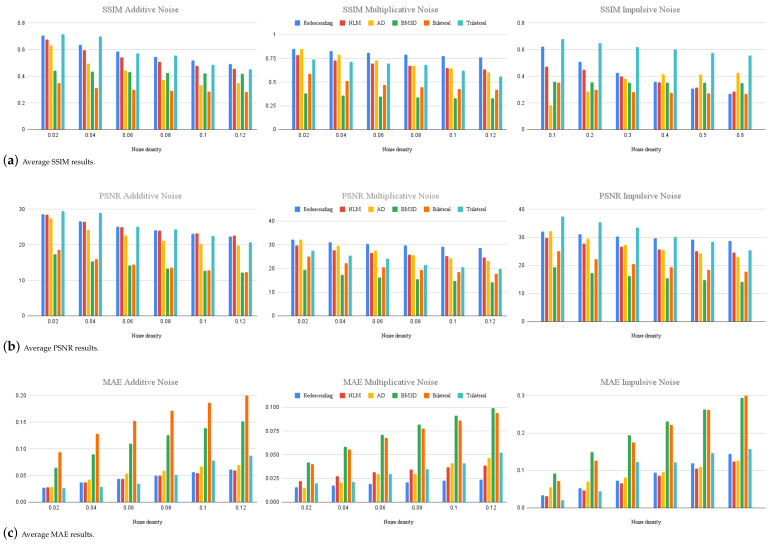
Graphical illustrations of the average results of the metrics for the four test images.

**Figure 4 entropy-25-01176-f004:**
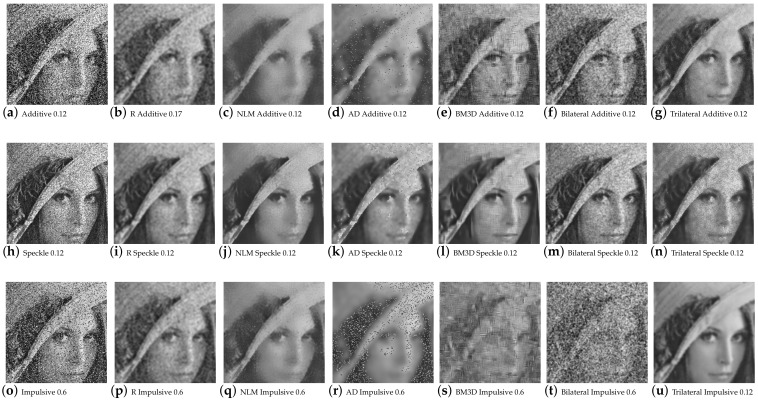
Qualitative results of the redescending (R), NLM, AD, BM3D, and bilateral Lena image.

**Figure 5 entropy-25-01176-f005:**
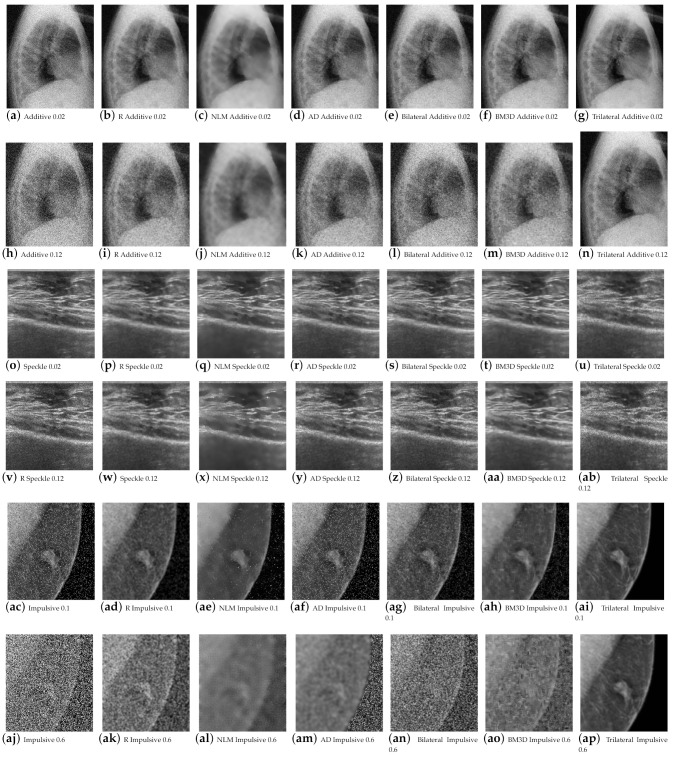
Results of redescending (R), NLM, and AD for the existing and proposed methods at low noise densities (additive = 0.02, multiplicative = 0.02, and impulsive = 0.1) and high noise densities (additive = 0.12, multiplicative = 0.12, and impulsive = 0.6).

**Figure 6 entropy-25-01176-f006:**
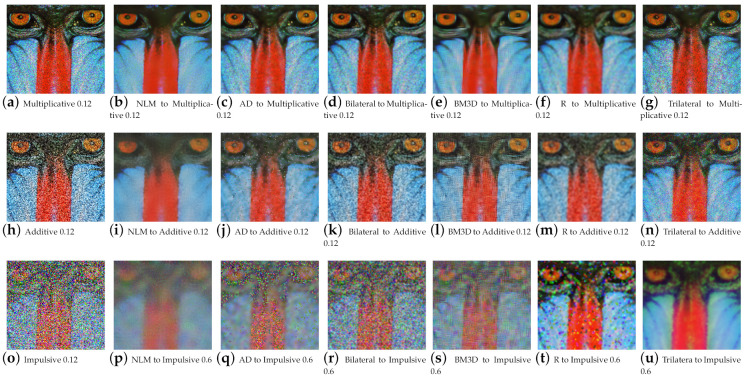
Qualitative results of the redescending (R), NLM, AD, BM3D, and bilateral for the baboon color image.

**Figure 7 entropy-25-01176-f007:**
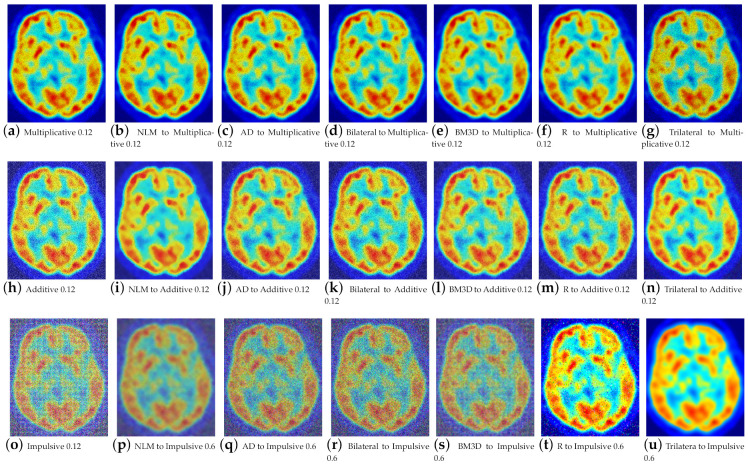
Results of filtering of the brain PET image at high densities.

**Figure 8 entropy-25-01176-f008:**
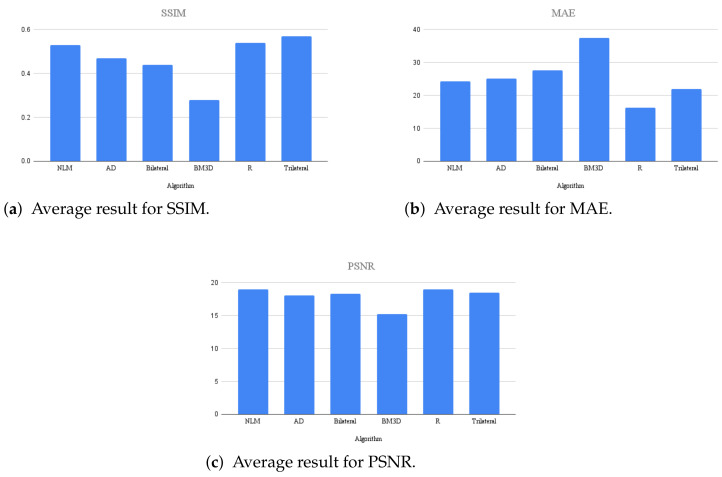
Average results for high noise densities in color datasets Kodak24 and CBSD68.

**Figure 9 entropy-25-01176-f009:**
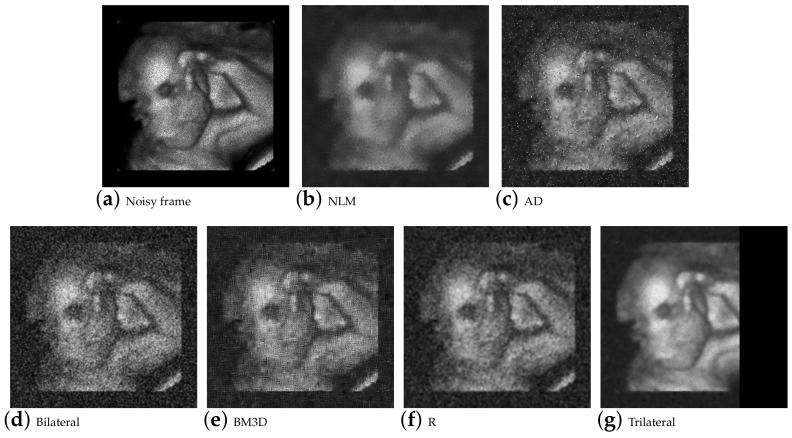
Results of the filtering of a frame from the video ultrasound.

**Figure 10 entropy-25-01176-f010:**
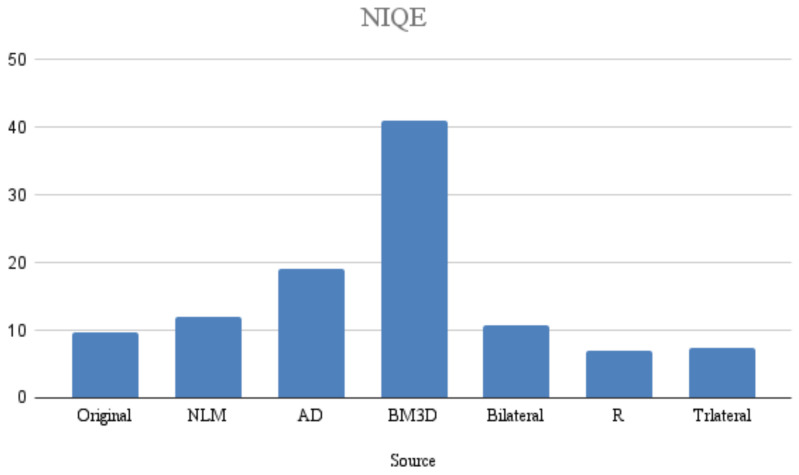
Graphical illustrations of the NIQE metrics.

**Table 1 entropy-25-01176-t001:** Noise models considered in this work [36].

Noise Type	Name	Expression
Additive	Gaussian	$\tilde{x}=\frac{1}{\sigma \sqrt{2\pi }}e^{\frac{-{\left(x-\mu \right)}^{2}}{2\sigma^{2}}}$
Multiplicative	Speckle	$\tilde{x}=x+nx,\;$ n=uniformlydistributedrandomnoise
Impulsive	Salt & Pepper	$\tilde{x}=\left\{\begin{array}{cc} 0 & \mathrm{with}\mathrm{probability}\;p/2\left(\mathrm{pepper}\right)\\ x_{i,j} & \mathrm{with}\mathrm{probability}\;p-1\\ 255 & \mathrm{with}\mathrm{probability}\;p/2\left(\mathrm{salt}\right) \end{array}\right.$

**Table 2 entropy-25-01176-t002:** Modified influence functions of the redescending M-estimator.

Influence Function	Formulae	Thresholds
Insha	$\psi_{I_{\left(r\right)}}\left(x\right)=x\cdot {\left[1+{\left(\frac{x}{c}\right)}^{4}\right]}^{2}\;for\;\left|x\right|\geq 0$	$c=k\cdot Med(|{\tilde{x}}_{i}-Med\left(h\right)|)$
Hampel’s three-part redescending	$\psi_{HAM_{\left(a,b,r\right)}}\left(x\right)=\left\{\begin{array}{cc} x & 0\leq |x|\leq a\\ a\cdot sgn(x) & a\leq |x|\leq r\\ a\cdot \frac{r-|x|}{r-b} & b\leq |x|\leq r\\ 0 & 0\leq |x| \end{array}\right.$	$r=k\cdot Med(|{\tilde{x}}_{i}-Med\left(h\right)|)$, a=0.15·r and b=0.85·r

**Table 3 entropy-25-01176-t003:** Restoration results in PSNR, SSIM, and MAE terms for Lena for the proposed method. Additive (A), multiplicative (M), impulsive (I), bilateral (Bi), and trilateral (Tri).

Noise Type	Noise Density	Denoising Method																	
		PSNR						SSIM						MAE					
		R	NLM	AD	BM3D	Bi	Tri	R	NLM	AD	BM3D	Bi	Tri	R	NLM	AD	BM3D	Bi	Tri
A	0.02	27.71	28.30	26.69	17.061	18.274	30.45	0.74	0.75	0.65	0.45	0.36	0.74	0.0227	0.0223	0.0208	0.0675	0.0980	0.0323
	0.04	26.37	26.34	24.18	15.088	15.622	29.542	0.67	0.66	0.54	0.44	0.32	0.66	0.0361	0.0348	0.0387	0.0936	0.1340	0.0505
	0.06	25.33	25.08	23.09	13.921	14.186	25.257	0.62	0.60	0.54	0.44	0.30	0.62	0.0415	0.0412	0.0468	0.1136	0.1593	0.0597
	0.08	24.48	24.16	22.28	13.123	13.250	24.514	0.59	0.56	0.50	0.43	0.30	0.57	0.0467	0.0470	0.0486	0.1298	0.1787	0.0682
	0.1	23.72	23.38	21.44	12.517	12.580	23.700	0.56	0.53	0.45	0.43	0.29	0.54	0.0511	0.0511	0.0514	0.1436	0.1944	0.0741
	0.12	22.99	22.77	21.00	12.030	12.072	23.011	0.53	0.50	0.51	0.42	0.29	0.48	0.0554	0.0563	0.0543	0.1559	0.2074	0.0816
M	0.02	29.01	30.24	30.46	18.433	23.460	23.425	0.81	0.79	0.82	0.38	0.51	0.79	0.0227	0.0223	0.0208	0.0538	0.0538	0.0323
	0.04	28.46	28.47	28.43	16.525	20.758	21.752	0.78	0.72	0.77	0.36	0.45	0.67	0.0254	0.0270	0.0279	0.0741	0.0737	0.0392
	0.06	27.95	27.59	27.31	15.397	19.167	20.276	0.75	0.68	0.72	0.35	0.41	0.64	0.0279	0.0303	0.0303	0.0893	0.0887	0.0439
	0.08	27.46	26.72	26.43	14.569	18.033	20.045	0.73	0.65	0.70	0.34	0.39	0.61	0.0301	0.0334	0.0324	0.1026	0.1015	0.0484
	0.1	27.04	26.06	25.72	13.936	17.161	19.873	0.71	0.62	0.69	0.34	0.37	0.58	0.0321	0.0357	0.0353	0.1139	0.1123	0.0518
	0.12	26.67	25.59	25.12	13.409	16.442	19.742	0.70	0.60	0.66	0.33	0.36	0.75	0.0339	0.0382	0.0363	0.1244	0.1221	0.0554
I	0.1	26.08	21.73	14.78	13.647	16.635	30.436	0.53	0.40	0.12	0.36	0.36	0.55	0.0347	0.0317	0.0572	0.0949	0.0733	0.0460
	0.2	21.96	22.65	13.74	11.503	13.713	28.456	0.50	0.37	0.24	0.35	0.31	0.54	0.0502	0.0471	0.0662	0.1509	0.1254	0.0683
	0.3	19.06	20.20	13.20	10.343	12.052	21.450	0.47	0.32	0.35	0.35	0.29	0.54	0.0658	0.0633	0.0766	0.1942	0.1718	0.0918
	0.4	16.77	17.75	12.33	9.474	10.835	18.562	0.45	0.29	0.37	0.35	0.28	0.49	0.0834	0.0782	0.0858	0.2327	0.2165	0.1134
	0.5	15.01	15.83	11.73	8.825	9.943	14.943	0.39	0.26	0.38	0.35	0.27	0.45	0.1015	0.0916	0.0929	0.2659	0.2566	0.1328
	0.6	13.47	14.17	12.10	8.306	9.202	14.202	0.35	0.24	0.36	0.34	0.27	0.35	0.1225	0.1055	0.1040	0.2954	0.2946	0.1530

**Table 4 entropy-25-01176-t004:** Average execution times (in seconds) with the increasing image size in the Lena image; bold indicates the fastest and underlined denotes the slowest.

Algorithm/Size	281.7 kB	436.1 kB	607.9 kB	783.9 kB
	768 × 768	1024 × 1024	1280 × 1024	1536 × 1024
Redescending	13.76	25.59	37.68	54.52
NLM	5.06	54.71	119.82	139.78
AD	20.36	45.83	60.68	91.43
BM3D	25.66	46.88	66.82	96.16
Bilateral	**0.09**	**0.143**	**0.22**	**0.31**
Trilateral	10.4	21.56	55.76	82.45

**Table 5 entropy-25-01176-t005:** Results of the different datasets with additive noise = 0.12, multiplicative noise = 0.12, and impulsive noise = 0.6.

Dataset	Noise	Algorithm	PSNR	SSIM	MAE	Dataset	PSNR	SSIM	MAE	Dataset	PSNR	SSIM	MAE
Mias	Additive	Redescending	18.43	0.57	0.0244	BSD68	21.37	0.88	0.017	DBUI	21.60	0.63	0.017
		NLM	18.65	0.58	0.0242		21.52	0.88	0.017		21.71	0.61	0.017
		AD	17.61	0.55	0.0251		18.47	0.78	0.019		18.74	0.43	0.020
		BM3D	13.33	0.38	0.0431		13.74	0.50	0.041		13.57	0.13	0.042
		Bilateral	18.23	0.60	0.0257		19.75	0.82	0.021		19.90	0.47	0.021
		Trilateral	18.11	0.51	0.020		18.01	0.77	0.016		22.07	0.59	0.017
	Multiplicative	Redescending	20.65	0.64	0.019		25.80	0.95	0.009		25.91	0.77	0.009
		NLM	20.59	0.71	0.018		23.96	0.92	0.011		23.07	0.68	0.012
		AD	20.14	0.68	0.018		22.17	0.89	0.012		22.11	0.69	0.012
		BM3D	18.70	0.57	0.021		19.64	0.82	0.018		20.77	0.57	0.016
		Bilateral	21.11	0.54	0.016		24.84	0.93	0.011		25.36	0.75	0.010
		Trilateral	22.98	0.63	0.018		21.45	0.79	0.022		22.7	0.65	0.014
	Impulsive	Redescending	16.41	0.51	0.026		18.70	0.80	0.016		15.76	0.33	0.035
		NLM	14.71	0.34	0.040		15.13	0.58	0.038		13.87	0.30	0.036
		AD	13.63	0.29	0.041		11.39	0.29	0.045		14.18	0.17	0.040
		BM3D	10.54	0.17	0.058		10.69	0.24	0.057		10.74	0.05	0.057
		Bilateral	13.66	0.30	0.043		13.89	0.43	0.042		18.81	0.45	0.017
		Trilateral	13.97	0.30	0.027		16.40	0.55	0.041		22.54	0.46	0.014

## Data Availability

Not applicable.

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
