# Peer review of "Enhancing Image Quality via Robust Noise Filtering Using Redescending M-Estimators"

_entropy, 2023, doi:10.3390/e25081176_

Round 1

Reviewer 1 Report (Previous Reviewer 1)

The paper has greatly improved. That said, it is very disappointing that authors were not able to remake a very old paper in the comparisons. Garnett et al.'s method can be easily implemented even in C++.

It is very limiting that a very small dataset for the color tests was employed. This paper leaves many answers unanswered for the reader.

The authors should answer at least:

1) If this method can be applied to HDR images.

2) Figure 6: log scale for the Y-axis, it is not clear what happens at the bottom. The authors should remove the scale-factor in the x-axis; the authors should place the actual Mpixel values. Why making difficult for the reader to understand the paper?

3) Is the code going to be released online upon acceptance?

The paper uses too many passive forms, English has passive but it should not be abused; especially when a subject of the action exists.

An example:

"The proposed filter was programmed in Matlab"

Who is the subject of this sentence?? This is a clear abuse of the passive form. The passive should be used when the subject of the action is totally unknown. The authors should have written:

"We wrote/programmed the proposed filter in Matlab".

Please, writing in the passive form in English makes the whole manuscript difficult to read and unnatural in the English language.

Author Response

Reviewer 2 Report (New Reviewer)

1. The novelty of the proposed method is not clear. The improvement over previous methods is not clarified.

 2. The introduction and comparison with the related work are incomplete. For example, the optimization-based sparse and low-rank models for image denoising are not included.

 3. Why do you handle the impulsive, additive, and multiplicative noise? There are many other types of noise. The authors need to clarify the motivation and orientation more clearly.

 4. The experiment results especially the competing methods are not convincing. The latest comparison methods are 2018 or before. I would suggest the authors compare with more state-of-the-art methods [1, 2]

[1] Two-stage convolutional neural network for medical noise removal via image decomposition, TIM, 2020.

[2] Multi-stage image denoising with the wavelet transform, PR, 2023.

must be improved

Author Response

Reviewer 3 Report (New Reviewer)

In this paper, the Authors propose a new denoising filter for dealing with all kind of most frequent noise within images (additive, multiplicative annd salt-pepper like). The method relies on the well-knowm Wiener filter embedded with a adaptive kernel.

The paper is interesting and have a fine experimental setup. However, there are some concerns:

- (this reviewer is an expert in image denoising". It is not usual to deal with "total" filters, in the sense to attack simultaneously different kind of noise. Note that the noise is well studied and when some data has gaussian noise, it means that the rest of components can be ignored. So, justify your approach soundly, otherwise, it does not make sense for real cases.

- Consider to reduce the number of plots dealing with computational cost. Note that just to indicate the average cost is enough and, however, analyze the computational complexity makes more sense.

- it is not fair to  compare results with filters that were designed for a specific noise. For instance, most of the filters used are specific for gaussian noise, so it does not make sense to compare with them when applied to multiplicative noise. Note that most of those filters have a modified version for other kind of noise. So, please, make the comparison in fair conditions.

- Additionally, explain the figure, "1.4823", for the sake of better understanting the paper.

Suggestion:

- Write always "Wiener", and no "wiener".

Although english grammar is in general fine,  there are many "typos": read carefully the manuscript and correct them all. Some examples:

- line 161, "bileteral"

- Algorithm, line 7, "is write".

There are many more.

Round 2

Reviewer 2 Report (New Reviewer)

1. In the deep learning era, it is natural to ask what the advantage of the proposed method over the learning-based network is. The authors should clarify the motivation of the proposed method more clearly.

2. The proposed work aims at various random noises, such as impulsive, additive, and multiplicative noise. Have you ever considered the non-iid noise, such as the stripe noise [1] and ringing artifacts noise [2]?

 3. The introduction of the related work for each category is too simple. The authors should provide more related image denoising works [1, 2]. Moreover, the introduction of the related methods of each category is not clear. Simply listing the existing work is not informative enough. I would like to see the authors sort out a clearer development of each category, namely the relationship between each related work including their advantage and disadvantage. This may be more informative for the readers.

[1] Remote sensing image stripe noise removal: from image decomposition perspective, TGRS, 2016.

[2] Two-stage convolutional neural network for medical noise removal via image decomposition, TIM, 2019.

 4. The experimental results are not convincing. On one hand, the number of testing data is too small. The authors need to report the quantitative results on a large-scale dataset. On the other hand, the competing methods are too old. It is recommended to add more recent SOTA competing denoising methods for a fair comparison. 

can be improved

Author Response

Reviewer 3 Report (New Reviewer)

The Authors did not include some of my suggestions but gave reasons for that.

Author Response

The reviewer did not add any further comments. We greatly appreciate their work to improve this article throughout this time.

Cordially,
Ángel Arturo Rendón Castro.

Round 3

Reviewer 2 Report (New Reviewer)

Thanks for the effort of the authors. They have partially resolved my concerns. However, there are several problems that make the manuscript unsatisfactory.

To be honest, the results are poor. There are obvious residual noise and the image content has been unexpectedly smoothed. I do not think the state-of-the-art denoising methods would bring these obvious artifacts. Moreover, no real complex noise images are provided. 

The motivation for the proposed methods is still unconvincing. Slow training is the intrinsic issue of deep learning, not denoising-based methods. Moreover, the inference time is extremely fast.

none

Author Response

This manuscript is a resubmission of an earlier submission. The following is a list of the peer review reports and author responses from that submission.

Round 1

Reviewer 1 Report

The paper seems to be incremental compared to previous work by the authors.

It is not clear if this filter can be applied to color images.

Quantitive results are not applied to large datasets but only on a few images.

The text needs a review by a native speaker or a grammar software (e.g., free version of grammarly). Moreover, there is Spanish in Figure 4, 6, etc.

Why not citing and testing BM3D or the bilateral filter (see Garnett version)?

It is not clear the timing, what is the computational complexity by varying the image size? A graph with varying image size should be added. Can the method be implemented on the GPU (this should be mandatory)? Can the method be applied to videos?

Missing comparisons with filters:

https://webpages.tuni.fi/foi/GCF-BM3D/

https://link.springer.com/article/10.1007/s11390-010-9351-z

https://ieeexplore.ieee.org/document/1518940

https://onlinelibrary.wiley.com/doi/abs/10.1111/j.1467-8659.2011.02078.x

Author Response

Comments are addressed in the attached document.

Reviewer 2 Report

This paper proposed an image denoising method based on redescending M-Estimators in terms of wiener filter, so as to suppress impulsive, additive and multiplicative noise in different densities. Some experiments are conducted to show the effectiveness of the proposed method.

-----------------------------------------------------

Comments to address:

1. Experiments are conducted on several images instead of a standard large benchmark. It is highly recommended to do so for strengthening the experiments.

2. Some recent image denoising methods are totally ignored, which should be included as related works for a discussion and even for experimental comparison if possible and appropriated. Some missing closed-related works are listed belows:

[a] Image denoising via sequential ensemble learning. TIP 2020. (Using a cascade of estimators for denoising)

[b] Image denoising using complex-valued deep CNN. Pattern Recognition. 2021. (Using a deep complex-valued estimator for denoising)

[c] Self2Self with dropout: Learning self-supervised denoising from single image. CVPR 2020. (Using a self-supervised estimator for denoising).

Above are only some missing references. The authors should also try to find other recent closely-related works for a comprehensive review.

3. Writing needs a big improvement. Current version is far from satisfactory, e.g., missing subject in the first sentence of the first paragraph of Section 2.1,

just to list few. Please have a careful reading and check.

4. A separate Related Work section with a comprehensive literature reivew is welcome.

Author Response

(The authors gave the same response as above.)

Round 2

Reviewer 1 Report

The paper has improved but Garnett et al.'s work is important work to be compared and the authors should look into this work as comparison.

Furthermore, the fact the method does not handle color images is very limiting. Nowdays, it is expected that a filter works for color images; at least to be applied for channel or in the CIELab color space to L only. Can this method work on HDR grayscale images? Or does it handle only 8-bit images?

Reviewer 2 Report

The revision has addressed my comments. It would be better to have a minor revision for language improvement.